# How Antimicrobial Resistance Is Linked to Climate Change: An Overview of Two Intertwined Global Challenges

**DOI:** 10.3390/ijerph20031681

**Published:** 2023-01-17

**Authors:** Roberta Magnano San Lio, Giuliana Favara, Andrea Maugeri, Martina Barchitta, Antonella Agodi

**Affiliations:** Department of Medical and Surgical Sciences and Advanced Technologies “GF Ingrassia”, University of Catania, 95123 Catania, Italy

**Keywords:** antimicrobial resistance, antibiotics, climate change, environment, public health, One Health approach

## Abstract

Globally, antimicrobial resistance (AMR) and climate change (CC) are two of the top health emergencies, and can be considered as two interlinked public health priorities. The complex commonalities between AMR and CC should be deeply investigated in a One Health perspective. Here, we provided an overview of the current knowledge about the relationship between AMR and CC. Overall, the studies included pointed out the need for applying a systemic approach to planetary health. Firstly, CC increasingly brings humans and animals into contact, leading to outbreaks of zoonotic and vector-borne diseases with pandemic potential. Although it is well-established that antimicrobial use in human, animal and environmental sectors is one of the main drivers of AMR, the COVID-19 pandemic is exacerbating the current scenario, by influencing the use of antibiotics, personal protective equipment, and biocides. This also results in higher concentrations of contaminants (e.g., microplastics) in natural water bodies, which cannot be completely removed from wastewater treatment plants, and which could sustain the AMR spread. Our overview underlined the lack of studies on the direct relationship between AMR and CC, and encouraged further research to investigate the multiple aspects involved, and its effect on human health.

## 1. Introduction

Antimicrobial resistance (AMR) is one of the most pressing health challenges globally, accounting for 670,000 annual infections in the European Union and nearly 33,000 related deaths [1]. While antimicrobial use exerts an ecological pressure on bacteria, poor infection prevention and control (IPC) practices favors their further spread. Bacteria could acquire multiple resistance mechanisms, which severely limit all the available treatments and sustain the quick spread of these multidrug resistant (MDR) organisms in the environment [2]. For this reason, the prudent use of antimicrobials, as well as an improvement of IPC in healthcare settings, are needed to get an effective response against AMR. In this scenario, the current climate crisis may make the already serious AMR issue worse. In fact, climate change (CC) is affecting human health both directly and indirectly, with an worrying soaring expected during the current century [3,4,5]. Direct effects include the rising temperatures, the increasing number of heat-related mortality and morbidity [6,7], and a more common occurrence of high-intensity storms [3,8,9]. On the other hand, indirect effects of CC on human health occur, affecting different environmental conditions, such as water quality and quantity [10,11,12,13], food and nutritional security [14,15], shifting ecosystems [16,17,18] and distribution of vectors [19,20]. These changes result in increasing waterborne disease, higher risk of foodborne and vector-borne diseases, and the potential increases in negative mental health outcomes [8,9]. According to the World Health Organization (WHO), CC is the single biggest health threat facing humanity, with an expected 250,000 additional deaths per year between 2030 and 2050 [21]. Many diseases are climate-sensitive, and the above-mentioned changes may lead to an increase in the spread of many bacterial and vector-borne diseases in humans, animals and plants [2]. In addition, the high occurrence of diseases described earlier could further increase the improper use of antimicrobials [22,23,24,25,26,27,28,29,30]. In order to reduce the burden of infectious diseases and AMR, therefore, the contemporary protection of humans, animals and the environment is necessary, in a perspective defined by the so-called One Health approach [31]. As proof of this, just think of the SARS-CoV-2 that has originated from the interplay between humans and animals: bats, which are vulnerable to global warming [32], have probably transmitted the virus to humans through pangolins as intermediate sources [33].

The resulting pandemic has only further underlined the need for antimicrobial stewardship to prevent AMR [34]. In fact, the pandemic made the AMR crisis worse, due to increasing antibiotic use and drug-resistant secondary infections in hospital [35,36]. Since CC and AMR have been exacerbated by human activities, there is the need to take actions and to contain their damages. However, it is necessary to understand that these two issues are not uniformly or equitably distributed, with higher and more significant impacts on low-medium income countries, which do not yet have funded action plans against AMR [37]. Aside from low socio-economic status, individuals with pre-existing health conditions and with close connections to the local environment often experience higher burden due to CC [3,38]. Indeed, the climate impact on human health also depends on the individuals’ sensitivity and adaptive capacity, which, in turn, are underpinned by social determinants of health. Although the potential link between AMR and CC is now established, their complex relationships and interactions have yet to be clarified. In fact, current evidence is still lacking and controversial, suggesting the need for ad-hoc surveillance and multidisciplinary research. To sum up what has been done so far, here we provide an overview of the current knowledge on these global challenges and their weaving. To do that, we searched PubMed and Web of Science databases for articles published from inception to 1 September 2022, using the following combination of terms: “Climate change” AND (“Antimicrobial resistance” OR “Antibiotic resistance”). After duplicates were removed, a total of 346 articles were obtained from literature databases and additional studies were identified from their reference lists. For the study selection, we did not apply specific inclusion/exclusion criteria in terms of year of publication, study design, settings, target of study and/or specific aspects of the relationship under consideration. Due to the general aim of this overview, the studies were selected if they discussed the potential relationship between climate change and antimicrobial resistance.

## 2. Needs for a Global Solution against AMR

The high consumption of antibiotics in the community, and in all healthcare settings, is undeniably the main factor contributing to AMR, especially in high-income countries [39]. This evidence increased the cost of research on antibiotics and the need to make them with a broader spectrum. Yet, even developing low-income and middle-income countries are seeing grown antibiotic use, rates of hospitalization and prevalence of healthcare associated infections (HAIs) [40]. Although the causes of AMR are varied and complex, they mainly arise as a consequence of mutations in bacteria and the selection pressure exerted by antibiotics that provide a competitive advantage for mutated strains. The emergence and spread of MDR pathogens continue to threaten the ability to deal with many of the most common infections. For instance, resistance rates to β-lactam antibiotics have ten-times increased since 1990 [41]. In healthcare settings, the spread of MDR organisms—such as *Enterobacteriaceae* carbapenem-resistant and methicillin-resistant *Staphylococcus aureus* (MRSA)—can be rapid, with severe consequences for vulnerable patients [34]. In line with this, mortality for AMR is also rising in developing countries, especially for newborns with maternal-acquired neonatal infections [42].

Globally, AMR is associated with longer state of illness, higher mortality rates, increased costs, and ineffective antibiotic treatments. In fact, patients with AMR infections show longer hospitalization and higher treatment costs than those infected by drug-susceptible strains [43,44]. The economic burden of AMR is also aggravated by the inability of performing surgeries or therapies (i.e., chemotherapy) in absence of effective antibiotics [45]. The global and rapid spread of multi- and pan-resistant bacteria, also known as “superbugs”, is especially alarming, because they cause infections that are not treatable with existing antibiotics. Moreover, novel and more effective antibiotics are not available for all countries. In many low- and middle-income countries, for example, people not only do not have full access to novel antibiotics, which are often more expensive, but also they often use common antibiotics without prescription [46]. The situation is aggravated by the uncontrolled antibiotic use in non-clinical fields (e.g., agriculture, aquaculture and intensive farming), which accounts for a volume four-times higher than that observed in humans [47].

In this scenario, several comprehensive strategies should be implemented to control AMR. Surveillance of antibiotic use, detection of resistance in human beings and animals, awareness campaigns on the correct use of antibiotic, and antibiotic stewardship programs in healthcare settings are just some of the interventions that countries should address [27,48]. Cooperation between prescribers, dispensers and patients is necessary to obtain a significant reduction of AMR. Moreover, the need for innovative strategies to tackle AMR should require investment both in the development of new antibiotics and in complementary technologies (e.g., vaccines) [49,50,51,52].

What is described in this paragraph about AMR points out similarities with CC [53]; for this reason, they are considered two intertwined challenges for Public Health. Not only that, but actions from single countries may also have potential effects on other countries and improve the situation globally; for this reason, global and national contributions are urgently required today more than ever.

## 3. Antimicrobial Resistance and Climate Change: The One Health Approach

According to the WHO, AMR is one of the top 10 global public health threats facing humanity, with misuse and overuse of antimicrobials as the main drivers for the development of drug-resistant pathogens. Although AMR is a natural phenomenon, the main drivers of its development and spread are “man-made”, including misuse and overuse of antimicrobials in humans, animals and plants [54]. Findings from studies demonstrating through whole genome sequencing the animal-to-human transfer of resistance genes [55], confirm the need of international actions tackling AMR across multiple sectors (i.e., human healthcare, agriculture and environment) [36].

Due to its broad scale, CC has been often compared to AMR; but these two issues are even more intertwined, since CC could indirectly contribute to AMR [53]. For this reason, it makes sense to address them through the One Health approach, focusing on the simultaneous protection of humans, animals and the environment. The relationship between global warming and infectious pathogens dates back many years [56], as suggested by many zoonotic microorganisms that have reached the human species [57]. For instance, the adaptation of bacteria—such as *Campylobacter*, *Salmonella* and *Vibrio cholerae*—to warmer temperatures have led to the re-emergence of these infectious diseases with the increasing temperatures in water systems [58,59]. Thus, resistance rates for these bacteria are bound to increase, with consequentially limited options to treat AMR pathogens [58,60,61,62]. Given the already decreased effectiveness of many antibiotics, implications due to the re-emergence of infectious bacteria from CC remain unclear [34]. Therefore, an early identification of infectious disease related to CC is crucial to manage “outbreaks” in all settings [59]. Additionally, the high use of antimicrobials in livestock farms and agriculture and the resulting increase in AMR necessitate a strict collaboration between veterinarians and clinicians for preventing the animal-to-human transmission [63,64]. For example, mass vaccination of animals—which is feasible and cost-effective—could help reducing livestock-mediated zoonoses [65].

In the same way, intersectoral strategies may be effective also against the climate crisis, even if further efforts are still needed to understand how the One Health approach may be enough to manage CC [63]. The majority of studies has thus far been focused on one of these dimensions. De Jongh and colleagues proposed honeybees as a model organism to better investigate the interactions between environmental changes and AMR in the One Health perspective [66]. Environmental pollutants, warming temperatures and other factors related to CC could negatively affect honeybees’ health, increasing the honeybees-related diseases and decreasing the efficacy of antimicrobials in treating pathogens [67,68,69]. Although several studies revealed potential indirect links to AMR in honeybees, few of them directly linked specific pollutive variables (e.g., neonicotinoids) to AMR. In general, evidence of the relationship between AMR, CC and environmental pollutants on honeybees’ health is lacking, suggesting the need for interdisciplinary research with the One Health approach. For this reason, a deep investigation of the potential relationship between AMR and CC should be encouraged to implement the synergy between human, animal and environmental health partners.

## 4. Antimicrobial Resistance and Climate Change: Two Intertwined Global Challenges

As stated above, the effects of CC on human health is getting worse with time, as well as its impact on Public Health that affects all the physical, natural, social and behavioral dimensions [70]. This worsening is very similar to that observed for AMR [71]. Factors that potentially contribute to the relationship between CC and AMR are summarized in Figure 1. Understanding how AMR evolved alongside CC can provide insights to better design future efforts and interventions [36]; for this reason, it would be necessary to better understand: (i) the relationship between CC and AMR with a focus on humans, animals and the environment; (ii) current strategies and future interdisciplinary research on their interactions; (iii) the impact of financing, political advocacy and global actions; and (iv) the role of the COVID-19 pandemic in this alarming scenario.

### 4.1. The Relationship between Climate Change and Antimicrobial Resistance at Human, Animal and Environmental Levels

While temperatures rise as a consequence of the climate crisis, AMR is increasing in humans, animals, plants and the environment [72,73]. Increasingly higher temperatures in fact are intimately linked to AMR, because they are associated with increased bacterial growth rates [74] and horizontal gene transfer [75]. In line with this, the climate crisis may also be responsible for the spread of new and re-emerging pathogens (e.g., Candida auris [31,76,77], Plasmodium falciparum [78]), which could harbor new resistance mechanisms and determine an increased number of hospital admissions. By 2050, people at risk for vector-borne diseases will be rise to 500 million [79]. Globally, malaria is the most prevalent vector-borne disease, with temperature and humidity that are considered two important factors involved in its spread [56,80,81]. Heat and humidity are also considered two risk factors for the increasing rates of salmonellosis, which is also becoming increasingly antibiotic resistant [82]. Similarly, vector-borne infections typical of tropical climates—such as Zika, Chagas disease, dengue and chikungunya—are slowly migrating towards countries with warmer temperatures, even in winter months [83]. A certain seasonality has been documented for influenza and, more recently, for some bacterial infections; for instance, higher incidence of Gram-negative infections occurs during the warmer months [84,85], reflecting the optimal growth conditions for many Gram-negative bacteria at 32–36 °C. Similarly, bloodstream infections (BSI) [86,87,88,89,90,91,92,93,94,95], HAIs [84,85], intra-abdominal [96] and surgical site infections [97,98] are more frequent during summer. The observed seasonal variations may depend on changes in human interactions, microorganisms’ characteristics and environmental factors, including water consumption, food preparation [99], temperature [85,94,100] and humidity [94].

An interesting question is whether CC could explain—at least partially—differences in AMR proportions observed in Europe, a region characterized by countries with different healthcare systems. In general, increasing temperature and population density led to increased rates of AMR [73,101]. The cross-sectional study conducted by Kaba and colleagues in 30 European countries shows that carbapenem-resistant Pseudomonas aeruginosa (CRPA), Klebsiella pneumoniae (CRKP), multi-resistant Escherichia coli (MREC) and Methicillin-resistant Staphylococcus aureus (MRSA) are significantly associated with the warm-season change in temperature [101]. The authors identify several confounders of AMR by using socio-economic indicators that are different in a multi-country European landscape. Interestingly, corruption perception is considered as a major predictor that is associated with AMR [101,102]. For instance, its contribution to explain CRPA variance (i.e., 78% alongside with warmer temperatures) is higher than that of outpatient antimicrobial use.

It has also been demonstrated that warmer climates might affect heavy metals or biocides concentrations in soil and water, as well as their uptake by bacteria, triggering AMR by co-resistance mechanisms [103,104,105,106,107]. As stated by Kusi and colleagues, healthcare facilities, wastewater, agricultural settings, and foods are the major vehicles of AMR in surface waters, which may contain antibiotic residues, biocides and heavy metals. Although CC could be considered an additional determinant of AMR in the aquatic environment [108], its causal effect should be considered with caution [101].

Rising temperatures are closely linked with flooding, population displacement and overcrowding caused by storms and precipitations [109]. These events lead to an increase in waterborne infections [110,111,112,113], which result in increased pressure on health systems and in a general collapse of sanitation infrastructures [114,115]. For instance, several waterborne diseases—such as cholera outbreaks in Haiti [116] and Nepal [117] after the earthquakes—are well-documented.

In addition, extreme weather events could lead to drought that, along with food scarcity and decline in the healthcare system, determine higher risk of acquiring antibiotic-resistant enteric pathogens [118]. Interestingly, diarrheal diseases such as Campylobacter, Salmonella and cholera survive better in warmer temperatures, explaining the recent re-emergence of these diarrheal infections [58,59]. For instance, as reported by Gudipati and colleagues, numerous outbreaks of S. typhi with AMR are well-known [31]. Furthermore, ensuring the high quality of water globally, in terms of microbial contamination, is an important problem to be addressed [119]. However, as reported by the Sustainable Development Goal (SDG) Synthesis Report on Water and Sanitation, the global SDG 6 targets by 2030 are far from being achieved; for this reason, it should be necessary to identify strategic opportunities for reducing diarrheal disease risks [120,121]. To do that, epidemiological studies have been developed to determine threshold values for water concentrations of E. coli, used as bacterial indicator species for fecal contamination [122,123]. However, further efforts are needed to improve the quality of water and contain the spread of common infections. Findings described above are summarized in Table 1.

Storms and precipitations can also damage wastewater and sewage infrastructure, increasing the risks of floodwater pollution [124]. Notably, wastewater is a reservoir for antibiotic-resistance genes (ARGs) [125], which have been identified as emerging pollutants both in soil [126] and water [127]. AMR bacteria and ARGs are not totally removed during wastewater treatment, and are discharged into the receiving environment [128]. Their detection and characterization in different matrixes and settings have been supported by the molecular methods, quantitative PCR and metagenomic analysis [129,130,131,132,133]. Nowadays, the characterization of the environmental resistome allows us to quantify and assess ARGs, with sewage and wastewater treatment plants (WWTPs) as the main objects of international monitoring studies. For instance, Zhang and colleagues used metagenomic sequencing to characterize ARGs and biocide resistance genes in four WWTPs in India [134]. Specifically, one WWTP treats hospital wastewater, a hotspot of resistome, due to the presence of antibiotic compounds and patients’ excrement. The other three WWTPs exclusively or partially receive municipal wastewater. Against expectations, municipal wastewater could be considered as a major vital source for ARGs, suggesting the alarming antibiotic use in the community. Moreover, the abundance of ARGs related to tetracycline and macrolide-lincosamide-streptogramin reflects the abuse of these antibiotics for the treatment of livestock and aquaculture [135].

In this scenario, microplastics are defined as conventional plastic with dimension <5 mm, accumulated in landfills or in the natural environment, that fragment rather than decompose. In particular, their accumulation has been widely reported in waters, soils and air [136]. Microplastics have been proposed as hotspots of horizontal gene transfer in water sources and crucial factors for the evolution of environmental microbial [137,138,139,140]. For this reason, metagenomic sequencing methods could assess the effects of microplastics on the spread of ARGs in several environmental settings, including waters, soils and air [136]. In particular, with respect to soil, pollutants, such as nitrogen fertilizers and farm waste used in agriculture (e.g., manure and wastewater), have been shown to increase AMR levels [124].

It is therefore plausible to state that the impacts of CC will result in an increased use of antimicrobials in humans, animals and plants. As a result of heat stress, for instance, poultries are overtreated with antibiotics, affecting their egg production and growth rates [141]. Globally, the overuse of antibiotics in livestock and poultry production is most often attributed to a low compliance with antimicrobial stewardship [142,143]. However, industrial animal farming also contributes to the rise of AMR by expanding deforestation, which makes closer the contacts of humans with wild animals that are carriers of emerging zoonotic diseases [144,145]. Moreover, without shifts in global meat consumption, agriculture will consume the global carbon budget needed for keeping global temperature rises under 2 °C by 2050. For these reasons, tackling AMR requires multidisciplinary actions, including human healthcare, agriculture and environment. Urgent efforts are also required for the productive industrial activity of aquaculture—defined as “the farming of aquatic organisms including fish, mollusks, crustaceans, and aquatic plants”—which involves the antibiotic use for treatment and prophylaxis. In Mediterranean aquaculture facilities, the overuse of antibiotics leads to their higher levels in the surrounding sediments and water column [146]. As reported by several studies, aquaculture is a hotspot for ARGs [147,148,149]. For instance, Pepi and colleagues reported that *E. coli* resistant to fluoroquinolones originates from the hospital, but also from the aquatic, environment. This represents a major issue for both human and animal health, as well as a concern for the environment. With respect to the relationship between AMR and CC, antibiotic-related modifications in bacterial cells are similar to those caused by increased temperatures [150]. The AMR indices of aquaculture are related with those of human clinical bacteria, temperature and climate vulnerability of countries. In fact, countries with higher risk of CC will probably face increased risks related to AMR [151]. Since further increase in AMR may be experienced, innovative and sustainable solutions are needed through the One Health approach [72,151].

### 4.2. Current Strategies and Future Interdisciplinary Research on the Relationship between Antimicrobial Resistance and Climate Change

Current evidence about the effects of CC on AMR should be translated into a common vocabulary and a fruitful dialogue between political leaders, policymakers, media, and the general public. With respect to the public, climate activists are used to shock people by painting the problem as an imminent catastrophic event. However, it is unlikely that this kind of approach could sustain behavioral changes. Instead, people should be guided in the fight against AMR and CC by reducing the prescriptions of unnecessary antibiotics [152]. This is an issue that also involves clinicians, who need more guides and practical tools to reduce unnecessary prescribing [153]. Since the current evidence is relatively scarce, multidisciplinary research is encouraged to develop epidemiological studies in different settings [154]. While a clear link between rising temperatures and infectious rates is emerging, the interaction between CC and AMR remains complex. However, the complexity of the interaction and the heterogeneity of available data hamper the research. In this scenario, the application of artificial intelligence (AI) and machine learning (ML) could provide a different point of view from which the well-known problem can be tackled. Rodríguez-González and colleagues, for example, provide an overview of the current application of AI in public health and epidemiology, with a special focus on the impact of climate crisis and AMR in disease epidemiology [155]. As reported by the authors, the use of an AI and ML could improve the current evidence on this complex issue. However, a surprisingly low number of studies have been published, despite many opinions in favor of such integration. The main reasons could lie in the complexity of integrating heterogeneous data and the lack of valid and impartial validation procedures [155]. Nowadays, the main efforts in the field of CC and AMR are related to forecasting and surveillance: the first one is necessary for deploying prophylactic measures and limiting the spread of diseases; the second one for monitoring the spread of diseases. For instance, the use of ecological niche models and algorithms to predict the geographical relocation of fly vector species suggest that CC will increase the probability of leishmaniasis spread in North America, North of México and Texas [156]. Similarly, an AI-focused approach reveals that temperature-related variables are the most effective meteorological factors for forecasting the weekly number of infectious diarrhea episodes, whereas weekly average rainfall is the least effective information [157]. Therefore, disease surveillance should be implemented in the context of Big Data, as done for example by Manogaran and Lopez in their big data surveillance system that analyzes the relationship between CC and the Dengue disease [158]. Therefore, both direct and indirect effects of climate crisis on AMR should be deeply investigated using multidisciplinary and innovative approaches.

### 4.3. The Economic and Political Interventions to Counteract Antimicrobial Resistance and Climate Change

The psychological distance represents a key barrier to people’s engagement with global problems, leading to a wrong and sweetened perception of the consequences on human health. In the era of expanding essential access, reducing inappropriate antibiotic use is a difficult challenge. To encourage people to act on climate crisis or antibiotic resistance, policymakers need to develop novel ways to make the issue closer to reality [152]. A plausible explanation of this twisted perception lies on the fact that also the national and international action plans should propose a more effective initiative, focused on the intersection of these two crises. A growing interest in the establishment of an Intergovernmental Panel on Climate Change (IPCC) for AMR is well-known [159]. Worldwide, 117 countries have AMR National Action Plans approved by their governments [160], and are aligned with the Global Action Plan for AMR [161]. However, concomitant global challenges, such as climate crisis, may threaten current AMR mitigation efforts. In line with this, the Global Leaders Group on Antimicrobial Resistance underlines the need for a more high-level political advocacy to ensure that AMR is included in high-level discussions on the climate crisis [154]. Interestingly, most countries do not fully implement the plans developed in response to CC. As stated by WHO, only half of the 101 countries included have developed a national health and CC strategy, and only 10% of countries used the resources required [162]. For both CC and AMR, the returns on investing in containment are expected to far outweigh the costs [163]. Additional financing is also needed to incorporate linkages between AMR and CC into existing One Health strategies and initiatives. As suggested by the 2015 Paris Agreement under the United Nations Framework Convention on Climate Change, building a global collective action is also necessary to manage the AMR threat. Notably, a comparison has been proposed—based on six key points—between the Paris Climate Agreement and the existing global AMR efforts [164]. First, AMR needs a unifying target to benchmark global progress which, for climate crisis, is represented by keeping global temperature rise well below 2 °C. As has been done for the climate crisis, it is also necessary to focus on social, collective and economic transformation instead of emphasizing individual behaviors [165]. Thirdly, escalating commitments through national AMR action plans (e.g., determined contributions pledged, reviewed and ratcheted every 5 years) is required to improve current efforts [166]. Fourth, a multistakeholder forum on AMR could be useful for ensuring an ongoing and inclusive dialogue, as well as for shaping equitable goals and actions [167,168]. Moreover, ongoing AMR action would be best informed by a global scientific stock taking every 5 years, useful to guarantee that policies are constantly informed by the best available evidence [159]. Lastly, an international legal framework could generate progress on AMR by creating a sustainable system with the active contribution of countries [169]. The resulting challenges have several common points with those posed by CC. In this context, the prediction of the future spread of AMR is necessary to estimate the burden that could be prevented by interventions aimed to reduce antibiotic use and/or by the development of new antibiotics. However, designing the trajectory of AMR changes is complex, arising from the complexity of estimating the cost, which includes: specific pathogen prevalence; AMR mechanisms; the level of transmissibility and the burden of infections; health consequences (e.g., chronic diseases and longer hospital stays); and the available alternative treatments [170,171]. An interesting approach to anticipate the possible risk is represented by the “scenario planning”—defined as a type of foresight method that enables exploration of alternative future worlds. Lambraki and colleagues, for instance, aim to explore alternative futures and actions to successfully address AMR under a changing climate in 2050. The authors use Sweden as the case study. Specifically, experts from multiple sectors involved in the study construct three alternative futures for each of the two promising interventions: (i) taxation of antimicrobials at point of sale, and (ii) infection prevention measures. All the alternative futures identify necessary actions for long-term AMR mitigation success. While taxation of antimicrobials reveals low impact that creates inequities, in two of the three alternatives, the infection prevention measures are considered as key strategies for containing AMR in 2050, contributing for the achievement of the Sustainable Development Goals (SDGs). In this way, it could be possible to tackle inequities underpinning AMR and CC, with the perspective of a holistic approach in which AMR and CC will be considered as interlinked issues [172].

### 4.4. The Impact of COVID-19 Pandemic on Antimicrobial Resistance and Climate Change

Countries and health systems continue to deal with the health and socio-economic effects of the COVID-19 pandemic. The economic losses associated with climate crisis determine increased pressure on economies already modified by synergistic effects of the COVID-19 pandemic [4]. For instance, the coexistence of outbreak infections (e.g., dengue infection) and COVID-19 pandemic has a negative impact on the healthcare system [173]. Similarly, food security and availability are affected by CC and the COVID-19 pandemic, with increased prevalence of undernourishment in 2020 than in 2019 [4]. In this context, it has been reported that COVID-19 pandemic has reduced financing available for climate action—including the reduction of greenhouse gas emissions or air pollution. Some studies have reported the main effects of climate on COVID-19 outcomes, showing the impact of warmer temperatures on mortality, as well as the contribution of extreme weather events on cardiopulmonary morbidity—a common risk factor for poor COVID-19 outcomes [174,175,176,177]. For this reason, urgent action is needed to strengthen health-system resilience [4]. With the COVID-19 pandemic, an increased number of hospital-acquired infections was evident, with higher risk for AMR [178]. Although COVID-19 is a viral infection, it could clinically progress to bacterial pneumonia, requiring antibiotic administration. An approximately 50% of secondary bacterial infection occurred in COVID-19 patients [179]. However, the Infectious Diseases Society of America (IDSA) states that only 8% of COVID-19 patients with bacterial superinfections required antibiotics. Along with antibiotics, a broad spectrum of pharmaceutical products used to address the spread of the virus can be collectively referred to Pharmaceutical and Personal Care Products (PPCPs) [180]. Although the WHO have developed guidelines for the PPCP usage, lack of knowledge and the COVID-19-related panic can lead to adverse effect on health.

As stated above, the contamination of waters with the so-called “emerging contaminants” (e.g., biocides, disinfectants, pharmaceuticals, hormonal drugs and cosmetic products) [180] was a problem that already existed before the COVID-19 pandemic [181]. However, an increasing amount of these contaminants has been observed in water bodies all over the world [182,183]. If the overuse of disinfectants, for example, associated with several side effects in humans (e.g., shortness of breath, cough, sneezing, abdominal pain, diarrhea and vomiting) [184], the abuse of biocides results in decreasing antibiotic sensitivity [185] while increasing cross-resistance and co-resistance [186]. Thus, the pandemic did nothing more than aggravate this problem [187,188], due to an increased residual of antibiotics reaching WWTPs from pharmaceutical companies, healthcare and domestic settings. Similarly, the overuse of plastics in the COVID-19 era (e.g., facial and surgical mask) determines the build-up of microplastics, which represent a stable substrate for microbes and ARG exchange. For these reasons, urgent interventions are needed to evaluate the impact of COVID-19 on AMR and the climate crisis, and to contain the harmful effects that these global emergencies are causing. The promotion of Public Health strategies to reduce the spread of AMR is therefore recommended, considering the implications of COVID-19 and climate change, which are intimately linked to AMR [180].

## 5. Discussion

CC and AMR are two of the greatest threats currently facing the world. Both have been exacerbated by, and can be mitigated with, human actions. In the era of climate crisis, the burden of infectious disease could be still reduced focusing on One Health strategies, recognizing how human health is closely connected to animals and the environment [31]. As reported by “the 2022 report of the Lancet Countdown on health and climate change”, vulnerable individuals (e.g., people > 65 years and children < 1 year) are more prone to the detrimental effect of heatwave days, a risk that is also exacerbated by the concomitant COVID-19 pandemic [4]. At the same time, climate crisis is affecting the spread of infectious diseases, determining higher risk of emerging diseases and AMR [4]. Due to time scale of cause and effect in CC [189] and AMR [190], future generations will not enjoy the benefits of carbon and antibiotic consumption. However, the trajectories of CC and AMR are strongly dependent on population levels and densities [191]. Alongside with natural CC (e.g., in water and ecosystems) involved in the potential increased spread of AMR, there is the urgent need for evaluating antimicrobial stewardship principles to prevent AMR during the COVID-19 pandemic [192]. Proper resources and education should be improved for healthcare workers and, in general, for the entire population. In line with this, the imbalance in the clinician–patient relationship could be relieved by promoting better education and public information campaigns for patients [193,194]; for this reason, successful public health interventions are based on the cooperation between human, animal and environmental health partners. The need to apply a systemic approach to the health of the planet is well-recognized. Accordingly, planetary health—defined as “the health of human civilizations and the natural systems on which they depend”—represents a promising interdisciplinary approach to promote alternative solutions for a resilient future [195]. Further efforts are needed not only to investigate the relationship between environmental change and human health, but also to evaluate the political and socio-economic systems underpinning those effects.

## 6. Conclusions

The complex commonalities between AMR and CC should be deeply investigated in a One Health perspective. Our overview on the current knowledge about the relationship between AMR and CC pointed out the need for further research to deeply investigate these two intertwined global challenges. To do that, it should be clear that CC increasingly brings humans and animals into contact, leading to outbreaks of zoonotic and vector-borne diseases with pandemic potential. Moreover, it is well-established that antimicrobial use in human, animal and environmental sectors is one of the main drivers of AMR. The COVID-19 pandemic is exacerbating the current scenario, by influencing the use of antibiotics, personal protective equipment and biocides, also resulting in higher concentrations of contaminants in natural water bodies. For this reason, contaminants (e.g., microplastics) cannot be completely removed from wastewater treatment plants, sustaining the AMR spread. Thus, a planetary health approach could offer potential solutions to commonly tackle these important public health issues.

## Figures and Tables

**Figure 1 ijerph-20-01681-f001:**
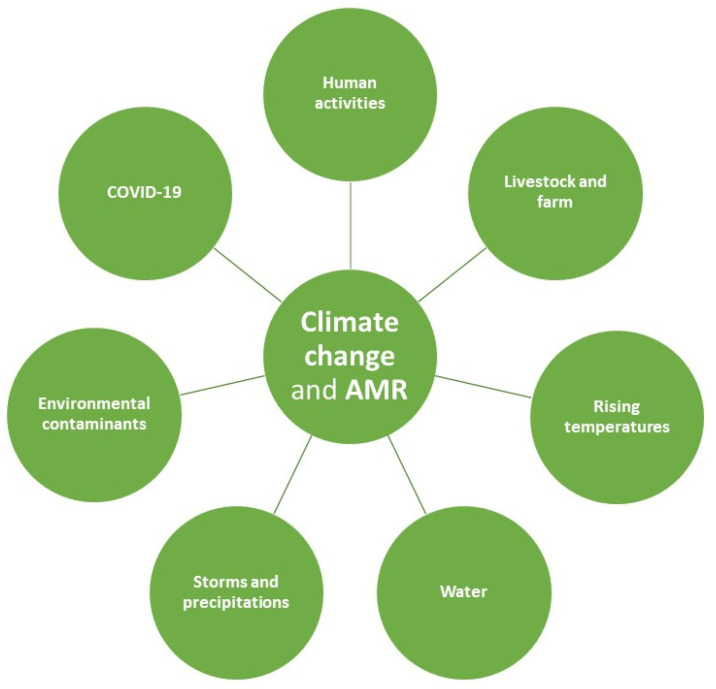
The main factors involved in the relationship between climate change and antimicrobial resistance. Abbreviation: AMR, Antimicrobial Resistance.

**Table 1 ijerph-20-01681-t001:** Summary of the relationship between climate change and infections.

Microorganisms	Role of Climate Change	Disease
*Campylobacter* spp. and *Salmonella* spp.	Rising temperatures in water system contributes to better survival of these microorganisms [31,58,59,82]	Waterborne and foodborne diseases
*Vibrio cholerae*	Rising temperatures led to natural disasters, determining better conditions for the microorganism survival [59,116]	Waterborne diseases (Cholera)
*Candida auris*	Gained thermotolerance and salinity tolerance on the wetland ecosystem [31,76,77]	Fungal infection (Candidiasis)
*Plasmodium falciparum*	Rising temperatures and humidity contributes to increased transmissibility [56,78,80]	Vector-borne disease (Malaria)
Zika, Chikungunya and Dengue viruses, *Tripanosoma cruzi*	Warmer temperatures led to rising spread of vectors, even in winter months [83]	Vector-borne diseases (Zika, Chikungunya, Dengue and Chagas diseases)
*Pseudomonas aeruginosa*, *Klebsiella pneumoniae*, *Escherichia coli*, and *Staphylococcus aureus*	Warm-season changes in temperature contributes to their optimal growth conditions at 32–36 °C [101]	Gram negative infections (especially in healthcare settings)
SARS-CoV-2	Increased aridity and prolonged droughts led to bats migration and increased viral transmission [32,33]	COVID-19 disease

## Data Availability

Not applicable.

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
