# Peer review of "How Antimicrobial Resistance Is Linked to Climate Change: An Overview of Two Intertwined Global Challenges"

_ijerph, 2023, doi:10.3390/ijerph20031681_

Round 1

Reviewer 1 Report

The authors base their study on the premise that antimicrobial resistance (AMR) and climate change (CC) are two health emergencies that can be considered public health priorities. Furthermore, they propose that a "one health" perspective should lead current investigations. The presented study provides an overview of the current knowledge about the AMR-CC relationship. In general, the authors include studies that indicate the need for a global health approach, e.g., CC favors human and animal contact, which induces outbreaks of zoonotic or vector-borne diseases with pandemic potential, and that antimicrobial use in the human and animal sectors is one of the main drivers of AMR. Moreover, they emphasize that the COVID-19 pandemic exacerbated the current scenario by influencing the use of antibiotics and biocides, leading to increased contamination of natural water bodies and, thus, the likelihood of AMR development. Finally, they highlight the lack of studies on the AMR-CC relationship and encourage further research on the matter and its direct effect on human health.

This review contributes to the field of antimicrobial resistance and climate change, two global health emergencies, highlighting the need for further research on their relationship to benefit human, animal, and environmental health. The manuscript appears professionally written, with minor errors amendable by detailed proofreading.

Given the above, the manuscript is endorsed for publication in "IJERPH".

Author Response

Dear Editor,

Please consider the revised version of the manuscript entitled “How Antimicrobial Resistance is linked to Climate Change: An Overview of Two Intertwined Global Challenges” in which we have considered all comments and suggestions from reviewers. This letter is intended for the convenience of the reviewers and contains the list of the requested changes. The following list of changes and answers to comments of Reviewer addresses all revisions made in the manuscript (in red font).

Reviewer 1

Comment: The authors base their study on the premise that antimicrobial resistance (AMR) and climate change (CC) are two health emergencies that can be considered public health priorities. Furthermore, they propose that a "one health" perspective should lead current investigations. The presented study provides an overview of the current knowledge about the AMR-CC relationship. In general, the authors include studies that indicate the need for a global health approach, e.g., CC favors human and animal contact, which induces outbreaks of zoonotic or vector-borne diseases with pandemic potential, and that antimicrobial use in the human and animal sectors is one of the main drivers of AMR. Moreover, they emphasize that the COVID-19 pandemic exacerbated the current scenario by influencing the use of antibiotics and biocides, leading to increased contamination of natural water bodies and, thus, the likelihood of AMR development. Finally, they highlight the lack of studies on the AMR-CC relationship and encourage further research on the matter and its direct effect on human health. This review contributes to the field of antimicrobial resistance and climate change, two global health emergencies, highlighting the need for further research on their relationship to benefit human, animal, and environmental health. The manuscript appears professionally written, with minor errors amendable by detailed proofreading.

Answer: We are grateful to Reviewer 1 for his/her comments.

Reviewer 2 Report

The current manuscript presents a quite interesting review of current knowledge on the link between antimicrobial resistance and climate change. Before acceptance, I advise that a few changes are made:

- The methodology section should not be an image, but a numbered section just like “1. Introduction”;

- What do you mean “No specific criteria for the study selection was applied”? Please clarify;

- Provide abbreviations for Figure 2, in this case AMR; also please provide the figure in higher resolution;

- A schematic representation of the most significant microbial infections that are linked to climate change should be done, also indicating the appointed reasons for that link;

- The conclusion section should be bigger, with a summary of the obtained results, future perspectives, etc., will further argumentation; part of the “4. Discussion” section can pass to the “5. Conclusion” section.

Author Response

Dear Editor,

Please consider the revised version of the manuscript entitled “How Antimicrobial Resistance is linked to Climate Change: An Overview of Two Intertwined Global Challenges” in which we have considered all comments and suggestions from reviewers. This letter is intended for the convenience of the reviewers and contains the list of the requested changes. The following list of changes and answers to comments of Reviewer addresses all revisions made in the manuscript (in red font).

Reviewer 2

The current manuscript presents a quite interesting review of current knowledge on the link between antimicrobial resistance and climate change. Before acceptance, I advise that a few changes are made:

C: The methodology section should not be an image, but a numbered section just like “1. Introduction”. What do you mean “No specific criteria for the study selection was applied”? Please clarify.

A: We are grateful to Reviewer 2 for his/her comments that helped us in improving our manuscript. As suggested, we removed the Figure 1 and described the methodology strategy at the end of the paragraph “Introduction”. We apologized if the selection criteria were confused. Please consider the revised version of our manuscript, in which we better describe the methodology used for the review.

C: Provide abbreviations for Figure 2, in this case AMR; also please provide the figure in higher resolution

A: Thank to Reviewer 2 for his/her comment. Please consider the revised version of the figure.

C: A schematic representation of the most significant microbial infections that are linked to climate change should be done, also indicating the appointed reasons for that link

A: Thank to Reviewer 2 for his/her comment. As suggested, we added a schematic representation (Table 1) of the relationship between climate change and infections.

C: The conclusion section should be bigger, with a summary of the obtained results, future perspectives, etc., will further argumentation; part of the “4. Discussion” section can pass to the “5. Conclusion” section.

A: We apologize if the conclusion was too short. Please consider the revised version of our manuscript, in which we summarized the main results of the paper and the future perspectives.

Reviewer 3 Report

In this paper titled 'How Antimicrobial Resistance is linked to Climate Change: An Overview of Two Intertwined Global Challenges' authors have provided a good discussion about the issue. I enjoyed reading it but I found some english language hick-ups. English could have been better. I have the following comments.    Line 72-73  As it is a review article, methodology is not required. Please remove it.    Line 218-219 "Rising temperatures are closely linked to the exponentially increase in storms and precipitations, which cause flooding, population displacement and overcrowding." Comments - Please correct english, also provide appropriate citation.   Line 229-231 "Globally, ensuring the high microbiological quality of water is an important problem to be addressed".  Comment - What do you mean by 'high microbiological quality of water'. Please clarify it.    Line 256-257 "In this scenario, microplastics have been proposed as hotspots of horizontal gene transfer in water sources and crucial factors for the evolution of environmental microbial  Comment - Please define microplastics briefly in next 1-2 sentences.    Line 434-436 "vulnerable individuals (e.g., people > 65 years and children < 1 year) are more prone to the detrimental effect of heatwave days, a risk that is also exacerbated by the concomitant COVID-19 pandemic." Comment - Please refer to the appropriate citation?   In line 45-46 - It should be 250,000.   Line 58-60,  "If on the one hand CC and AMR have been exacerbated by human activities, on the other hand there is room to repair the damages" Comment - Please correct the sentence.  

Author Response

Dear Editor,

Please consider the revised version of the manuscript entitled “How Antimicrobial Resistance is linked to Climate Change: An Overview of Two Intertwined Global Challenges” in which we have considered all comments and suggestions from reviewers. This letter is intended for the convenience of the reviewers and contains the list of the requested changes. The following list of changes and answers to comments of Reviewer addresses all revisions made in the manuscript (in red font).

Reviewer 3

C: In this paper titled 'How Antimicrobial Resistance is linked to Climate Change: An Overview of Two Intertwined Global Challenges' authors have provided a good discussion about the issue. I enjoyed reading it but I found some english language hick-ups. English could have been better. I have the following comments.   

A: We are grateful to Reviewer 3 for his/her comments that helped us in improving our manuscript.

C: Line 218-219 "Rising temperatures are closely linked to the exponentially increase in storms and precipitations, which cause flooding, population displacement and overcrowding." Comments - Please correct english, also provide appropriate citation.  

A: We apologize if the sentence was not clear. Please consider the revised version of our manuscript.

C: Line 229-231 "Globally, ensuring the high microbiological quality of water is an important problem to be addressed".  Comment - What do you mean by 'high microbiological quality of water'. Please clarify it.   

A: We apologize if the sentence was not clear. Please consider the revised version of our manuscript.

C: Line 256-257 "In this scenario, microplastics have been proposed as hotspots of horizontal gene transfer in water sources and crucial factors for the evolution of environmental microbial. Comment - Please define microplastics briefly in next 1-2 sentences.   

A: We thank the Reviewer 3 for the suggestion. Please consider the revised version of our manuscript.

C: Line 434-436 "vulnerable individuals (e.g., people > 65 years and children < 1 year) are more prone to the detrimental effect of heatwave days, a risk that is also exacerbated by the concomitant COVID-19 pandemic." Comment - Please refer to the appropriate citation?  

A: As suggested, we added the reference in the revised version of our manuscript.

C: In line 45-46 - It should be 250,000.  

A: We thank the Reviewer 3 for the suggestion. Please consider the revised version of our manuscript.

C: Line 58-60,  "If on the one hand CC and AMR have been exacerbated by human activities, on the other hand there is room to repair the damages" Comment - Please correct the sentence.  

A: We thank the Reviewer for the suggestion. Please consider the revised version of our manuscript.